# Full genome viral sequences inform patterns of SARS-CoV-2 spread into and within Israel

Danielle Miller[1,16], Michael A. Martin [2,3,16], Noam Harel[1,16], Omer Tirosh[1,16], Talia Kustin[1,16], Moran Meir[1], Nadav Sorek[4], Shiraz Gefen-Halevi[5], Sharon Amit[5], Olesya Vorontsov[6], Avraham Shaag[6], Dana Wolf[6], Avi Peretz[7,8], Yonat Shemer-Avni[9], Diana Roif-Kaminsky[10], Naama M. Kopelman[11], Amit Huppert[12,13], Katia Koelle[2,14] & Adi Stern [1,15✉]

Full genome sequences are increasingly used to track the geographic spread and transmission dynamics of viral pathogens. Here, with a focus on Israel, we sequence 212 SARS-CoV-2 sequences and use them to perform a comprehensive analysis to trace the origins and spread of the virus. We find that travelers returning from the United States of America significantly contributed to viral spread in Israel, more than their proportion in incoming infected travelers. Using phylodynamic analysis, we estimate that the basic reproduction number of the virus was initially around 2.5, dropping by more than two-thirds following the implementation of social distancing measures. We further report high levels of transmission heterogeneity in SARS-CoV-2 spread, with between 2-10% of infected individuals resulting in 80% of secondary infections. Overall, our findings demonstrate the effectiveness of social distancing measures for reducing viral spread.

[1] The Shmunis School of Biomedicine and Cancer Research, George S. Wise Faculty of Life Sciences, Tel Aviv University, Tel Aviv, Israel. [2] Department of Biology, Emory University, Atlanta, GA, USA. [3] Population Biology, Ecology, and Evolution Graduate Program, Laney Graduate School, Emory University, Atlanta, GA, USA. [4] Microbiology Laboratory, Assuta Ashdod University-Affiliated Hospital, Ashdod, Israel. [5] Clinical Microbiology Laboratory, Sheba Medical Center, Ramat-Gan, Israel. [6] Clinical Virology Unit, Hadassah Hebrew University Medical Center, Jerusalem, Israel. [7] The Azrieli Faculty of Medicine, Bar-Ilan University, Safed, Israel. [8] Clinical Microbiology Laboratory, The Baruch Padeh Medical Center, Poriya, Tiberias, Israel. [9] Clinical Virology Laboratory, Soroka Medical Center and the Faculty of Health Sciences, Ben-Gurion University of the Negev, Beer-Sheva, Israel. [10] Microbiology Division, Barzilai University Medical Center, Ashkelon, Israel. [11] Department of Computer Science, Holon Institute of Technology, Holon, Israel. [12] Bio-statistical and Bio-mathematical Unit, The Gertner Institute for Epidemiology and Health Policy Research, Chaim Sheba Medical Center, 52621 Tel Hashomer, Israel. [13] School of Public Health, The Sackler Faculty of Medicine, Tel-Aviv University, 69978 Tel Aviv, Israel. [14] Emory-UGA Center of Excellence of Influenza Research and Surveillance (CEIRS), Atlanta, GA, USA. [15] Edmond J. Safra Center for Bioinformatics, Tel Aviv University, Tel Aviv, Israel. [16] These authors contributed equally: Danielle Miller, Michael A. Martin, Noam Harel, Omer Tirosh, Talia Kustin. ✉email: sternadi@tauex.tau.ac.il

In December 2019, an outbreak of severe respiratory disease was identified in Wuhan, China[1]. Shortly later, the etiological agent of the disease was identified as severe acute respiratory syndrome coronavirus 2 (SARS-CoV-2)[2,3], and the disease caused by the virus was named coronavirus disease 19 (COVID-19). The virus has since spread rapidly across the globe, causing a WHO-declared pandemic with social and economic devastation in many regions of the world[4]. The infectious disease research community has quickly stepped up to the task of characterizing the virus and its replication dynamics, describing its pathogenesis, and tracking its movement through the human population. Parameterized epidemiological models have been particularly informative of how this virus has spread with and without control measures in place, e.g., ref. [5], and have been used to project viral spread both in the short-term[6] and in the more distant future[7].

Along with epidemiological analysis based on case reports and COVID-19 death data, sequencing of viral genomes has become a powerful tool in understanding and tracking the dynamics of infections[8,9]. So-called genomic epidemiology allows for effective reconstruction of viral geographical spread as well as estimation of key epidemiological quantities such as the basic reproduction number of a virus, its growth rate and doubling time, and patterns of disease incidence and prevalence. Such insights have been used to inform policy makers during various pathogen outbreaks, as occurred for example in the 2014–2016 outbreak of Ebola virus in West Africa[10,11] and during this current SARS-CoV-2 pandemic[12,13].

Here, we set out to sequence SARS-CoV-2 from samples across the state of Israel, with the aim of gaining a better understanding of introductions of the virus into Israel, spread of the virus inside the country, and the epidemiology of the disease, including (a) the basic reproduction number of the virus before and after social distancing measures were implemented, and (b) the extent of viral superspreading within Israel. As pointed out recently[14], caution should be exercised when interpreting viral dynamics based on genetic data only. We thus incorporated here extensive, high-resolution epidemiological data that exist regarding the outbreak in Israel.

The first confirmed cases of SARS-CoV-2 infection in Israel were reported in mid-February, followed by many identified SARS-CoV-2 cases in travelers returning to Israel mainly from Europe and the United States. Growth in the number of verified cases rapidly ensued, which led to increased measures of social distancing, including the cessation of passenger flights to Israel, school closure, and eventually a near complete lockdown across the entire state of Israel. Quarantining of returning travelers from Europe was implemented between February 26 and March 4, 2020, and subsequently all incoming travel to Israel (including from the U.S.) was arrested on March 9. In the meantime, the rate of testing was ramped up, eventually reaching a rate of more than 1500 tests per million people per day. The reported daily incidence and reported numbers of daily severe cases peaked around mid-April and dropped steadily up to the time of this manuscript's initial submission (May 2020). Despite this knowledge, many questions remain: Which of the multiple SARS-CoV-2 introductions resulted in sustained local transmission? How did the virus spread across the state? What was the magnitude of the virus's reproduction potential within Israel, and to what extent did control measures mitigate its spread through March–April?

Here, through a comprehensive set of phylogenetic and phylodynamic analyses, we quantitatively address these questions. We show that travelers from the U.S. contributed significantly more to viral spread as compared to their proportion in incoming infected travelers. We use a phylodynamic approach to estimate the basic reproduction number of the virus, and show a substantial reduction in viral spread following the implementation of social distancing measures. Finally, we report high levels of transmission heterogeneity in SARS-CoV-2 spread, with between 2% and 10% of infected individuals resulting in 80% of secondary infections.

## Results and discussion

To gain a better understanding of the dynamics of SARS-CoV-2 spread into and within Israel, we sequenced the virus from a cohort of patients representing a random sample across Israel, resulting in 212 full-genome SARS-CoV-2 sequences (Methods). A total of 224 unique single nucleotide variants (SNVs) were identified between the Wuhan reference sequence and this set of sequences from Israel. Figure 1 shows the distribution of identified SNVs along the genome and their counts in the sequenced samples. Of these SNVs, 141 were non-synonymous, 72 were synonymous, and the remaining 11 were in non-coding regions. One of the most abundantly detected SNVs was a non-synonymous variant D614G found in the spike protein, which

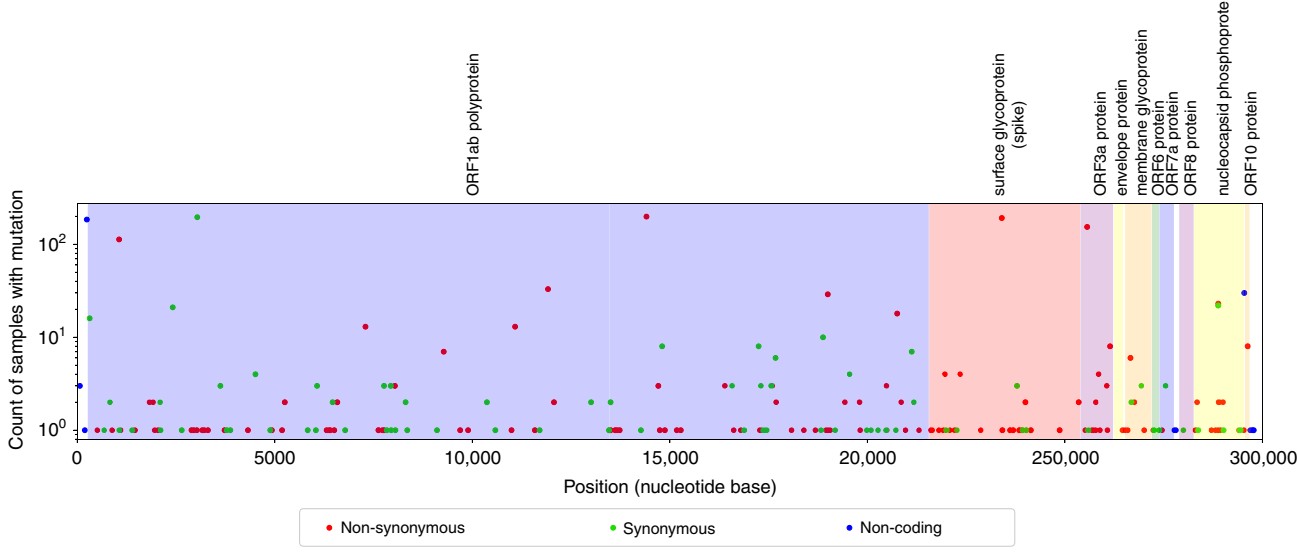

**Fig. 1 Variation found in sequenced samples from Israel.** The *x*-axis corresponds to the SARS-CoV-2 genome and the *y*-axis provides counts of identified SNVs across the viral genome. Source data are provided as a Source Data file.

**a**

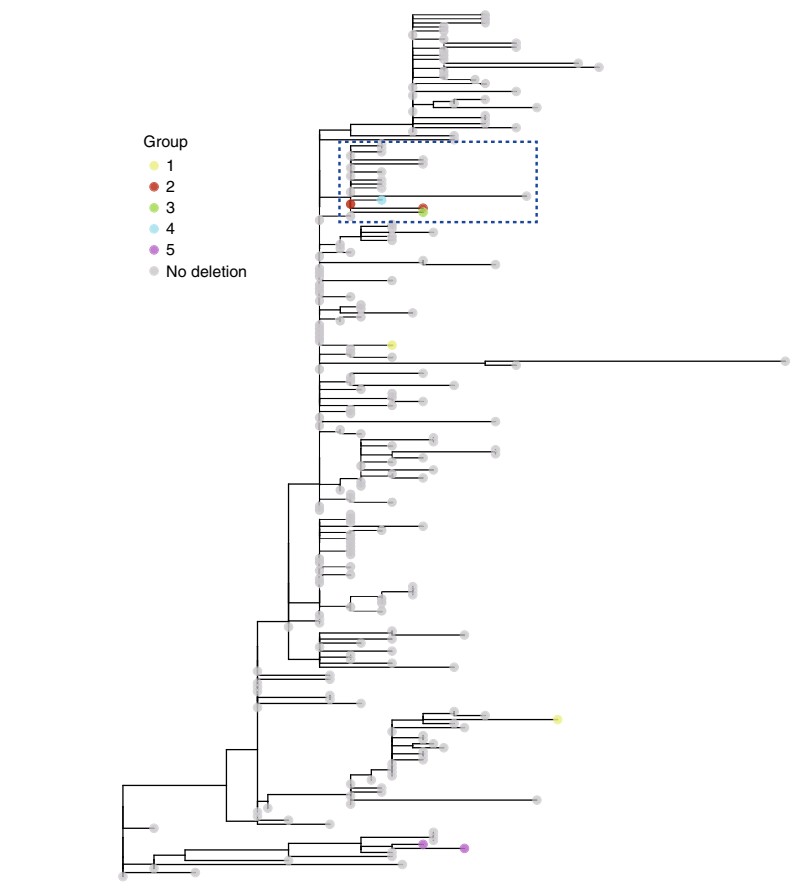

**b**

| # | Genome sites | Length (nt) | ORF/Genomic location | Putative effect | # samples found in | Sample IDs | Number of reads supporting deletion | Geographic location (district) | # non-Israeli samples found in refs.[1,2] |
|---|---|---|---|---|---|---|---|---|---|
| 1 | 686–694 | 9 | ORF1ab polyprotein | Deletion of 3 amino acids | 2 | , 2086008 130710157 | , 3575 1852 | Jerusalem, Tel Aviv | 222 |
| 2 | 3882–3899 | 18 | ORF1ab polyprotein | Deletion of 6 amino acids and an additional single amino acid mutation | 2 | ,2089839 2089852 | 605 ,427 | Jerusalem | -- |
| 3 | 27387-27396 | 10 | End of ORF6 and start of ORF7a | Stop codon of ORF6 is recreated. Start codon of ORF7a is deleted with no in-frame replacement | 1 | 13077726 | 3801 | Tel Aviv | -- |
| 4 | 28254 | 1 | End of ORF8 | Last amino acid is replaced by a 5 amino acid addition | 1 | 2086033 | 2849 | Jerusalem | 15 |
| 5 | 29746–29748 | 3 | 3' UTR | Non-coding, un-known | 2 | ,51137844 51141225 | 42,147 | South | -- |

[1]Based on >50,000 SARS-CoV-2 sequences downloaded from GISAID, July 17 2020, and also reported in https://virological.org/t/common-microdeletions-in-sars-cov-2-sequences/485
[2]We note that these are underestimates as deletions are sometimes masked in reported sequences.

**Fig. 2 Deletions found in Israeli samples. a** Maximum-likelihood tree of Israeli sequences highlighting sequences found with deletions. Sequences are color-coded by the groups described in **b**. A clade with three independent deletions occurring in four samples is boxed. Source data are provided as a Source Data file.

was present in 90% of the sequences. This variant has generated much interest as it has been reported to potentially increase the transmissibility of the virus[15]. However, additional analyses have suggested that the observed increase in this variant's frequency may be due to stochastic effects[16].

We also found five different high confidence genomic deletions, spanning between 1 and 18 nucleotides (Fig. 2) (Methods). Each of these deletions was found in one to two samples. Three of the five deletions occur in multiples of three and are in-frame deletions or affect non-coding regions. Of the remaining two

deletions, deletion #3 spans ten nucleotides, and likely prevents the translation of ORF7a. Deletion #4 occurs at the end of ORF8 and causes the replacement of the last amino acid with an additional five amino acids. Notably, an 81-nucleotide in-frame deletion in ORF7a has been previously reported[17], as has a 382-nucleotide deletion in ORF8 (ref. [18]), suggesting that the virus is to some extent tolerant to deletions in these ORFs.

When focusing on deletions that occurred in two samples, we noted that deletion #5 was present in two related samples that were sampled 5 days apart from each other. Deletion #1, on the other hand, appeared in two samples located in very remote clades of the phylogeny. This deletion has been observed multiple times in various sequences with diverse genetic backgrounds, including sequences from many different countries across the world, suggesting that it has arisen multiple independent times (Fig. 2b). Deletions #2, #3, and #4 revealed an intriguing pattern: three independent deletions (one of which was present in two samples) were all part of the same clade that included 18 samples (Fig. 2). One non-synonymous SNV defined this clade: S2430R in ORF1b, which affects the non-structural protein NSP16. This protein has been reported to be a 2′O-methyltransferase that enhances evasion of the innate immune system[19]. To follow up on our finding of deletions in this clade, we examined a set of over 50,000 global sequences, and found that 137 sequences likely belong to the clade defined by S2340R, with eight of these sequences bearing short deletions (Supplementary Table 1). Interestingly, we found that the proportion of these unique deletions observed in the S2340R-defined clade was 5%, significantly higher than the proportion of unique deletions observed across the entire global tree (1.8%) ($P = 0.01$; hypergeometric test), leading us to cautiously suggest that S2340R is associated with a higher rate of deletions.

While further in-depth investigation of SARS-CoV-2 indels is clearly needed, at this point we conjecture that the deletions we detected are neutral or to some extent deleterious, and that deletions in SARS-CoV-2 are likely to occur frequently given the number of deletions detected in our samples.

**Origins and transmission patterns in Israel**. We next set out to explore patterns of SARS-CoV-2 introduction into Israel. Figure 3 shows the time-resolved phylogeny inferred using 214 Israeli sequences (the 212 sequenced here and two additional ones sequenced previously) in addition to 4693 representative sequences from across the world. This phylogeny allowed us to characterize the major viral clades circulating within Israel and to infer the geographic sources and timing of virus introductions into the state. We found multiple introductions into Israel from both the U.S. and Europe, the latter including mainly the U.K., France, and Belgium. Over 70% of the clade introductions into Israel were inferred to have occurred from the U.S., while the remaining were mainly from Europe. To rule out that this result is due to biases in geographic sampling, we first note that in our sample, the number of sequences from Europe ($n = 1991$) was higher than the number from the U.S. ($n = 1195$). We further quantified sampling noise by bootstrapping the Israeli samples and exogenous samples, leading to confidence intervals ranging between 50% and 80% for U.S. clade importations (Methods). We noted considerably lower proportions of importations from the U.S. into other countries we examined (Supplementary Fig. 1), making it unlikely that our results arise from systematic biases of U.S. sequences relative to sequences from other regions. We validated the robustness of our inference to changes in underlying evolutionary model parameters and possible biases in ancestral state assignments, and found that any biases stemming from uncertainty in phylogenetic reconstruction are negligibly small

(Supplementary Fig. 2). Finally, we note that attribution of an Israeli sequence to a U.S. clade was normally based on two to four shared mutations, making it exceedingly unlikely that parallel independent substitutions occurred that could alternatively explain these patterns. However, we acknowledge that additional sequencing may change some of the inferences we make here, as has been shown elsewhere[20].

Throughout the epidemic in Israel, very close monitoring of all incoming infected travelers was imposed, and reports show that only ~27% of infected returning travelers were from the U.S. (Supplementary Fig. 3). There is a strong discrepancy between this 27% estimate and the 70% estimate for clade introductions, and this discrepancy holds even when considering our lower bound bootstrap estimate of 50% for clade importations (mentioned above). This suggests that the travelers returning from the U.S. contributed substantially more to the spread of the virus in Israel than would be proportionally expected. This may have occurred due to the gap in policy that allowed returning non-European travelers to avoid quarantine until March 9, or due to different contact patterns of those who returned from the U.S. Moreover, by examining the timing of viral importation events from the U.S. into Israel, we found that up to 55% of the transmission chains in Israel (118 out of 214; Methods) could have been prevented had flights from the U.S. been arrested at the same time that flights from Europe were arrested (between February 26 and March 4, instead of by March 9).

As the pandemic spread, entry into Israel was restricted, and local transmission became dominant. Transmission patterns into and between five geographical regions in Israel (North district, Tel Aviv district, South coast district, Jerusalem district, and South district) are shown in Fig. 4. While most transmission occurred inside defined regions, transmission between distinct regions was also observed, such as, for example, movement between Jerusalem and the north district of Israel.

**Phylodynamic modeling of viral spread in Israel**. To estimate the basic reproduction number of SARS-CoV-2 in Israel initially and then following the implementation of social distancing measures, we performed coalescent-based phylodynamic inference using the PhyDyn program implemented in BEAST2 (Methods). We note that existing phylodynamic analyses of SARS-CoV-2, focusing on a number of different geographic regions across the world, have shown that the effective reproduction number of the virus has decreased over time, as quarantine and social distancing measures have been implemented[21–23]. However, many of these analyses have to date modeled reductions in the reproduction number as stemming from the depletion of susceptible individuals[23], rather than from reductions in the basic reproduction number $R_0$, the latter of which would be consistent with lowering of contact rates. Other analyses, particularly those that use the birth–death model approach for phylodynamic inference, have allowed for changes in $R_0$ over time[21,22], but cannot as easily accommodate structure in the infected host population (e.g., that some individuals are exposed but not yet infectious, and that transmission heterogeneity exists between infected individuals). Our phylodynamic analysis here, based heavily on existing coalescent-based model structures that have been applied to SARS-CoV-2[24], instead allows for this structure to be accommodated and for $R_0$ to change in a piecewise fashion over time.

Our phylodynamic analysis assumes an underlying susceptible–exposed–infected–recovered (SEIR)-type epidemiological model for SARS-CoV-2 transmission dynamics and explicitly incorporates transmission heterogeneity (Supplementary Fig. 4, Methods). Recent epidemiological analyses have estimated considerable

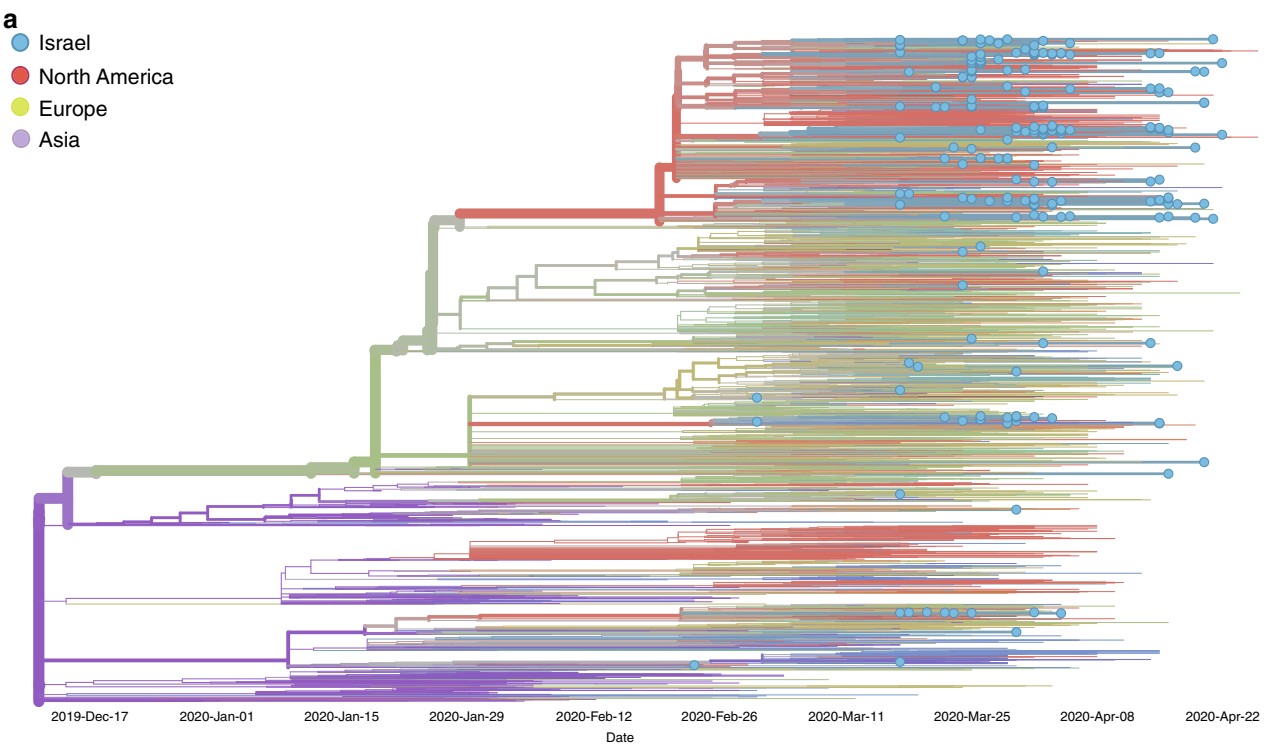

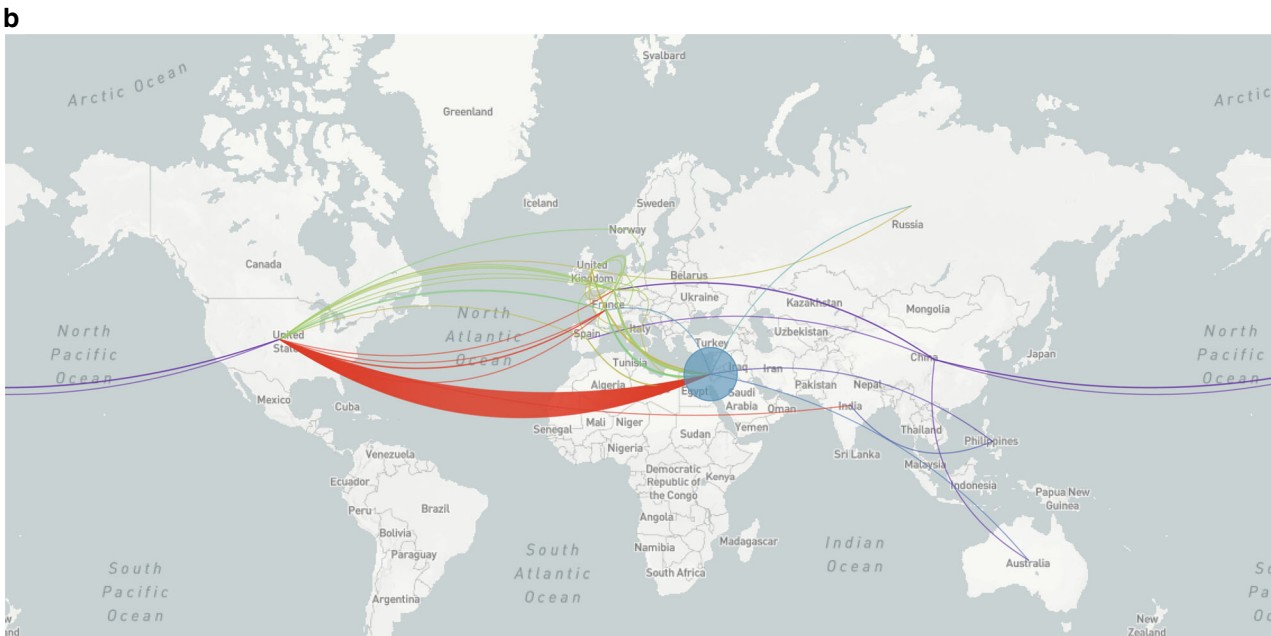

**Fig. 3 Patterns of SARS-CoV-2 introduction into Israel. a** Time-resolved phylogeny inferred using viral sequences from Israel (blue tips) and around the world (tips without dots). Lineages are colored by inferred region of circulation. Phylogeographic analysis reveals multiple introductions into Israel, mainly from the U.S. **b** Map of phylogenetically inferred introductions into Israel highlighting the dominance of the U.S. and to a lesser extent Europe as the geographic sources of SARS-CoV-2 introductions into Israel. Figure (including map) generated using NextStrain[32]. Source data are provided as a Source Data file.

levels of SARS-CoV-2 transmission heterogeneity, with ~7–10% of infected individuals estimated to be responsible for 80% of secondary infections[25,26]. Instead of assuming a given level of transmission heterogeneity for Israel, we instead performed phylodynamic inference for the SEIR model across a range of transmission heterogeneities. Specifically, the SEIR-type model implemented two classes of infectious individuals, corresponding to a highly infectious subset of individuals ($I_h$) and a less

infectious subset of individuals ($I_l$). Exposed ($E$) individuals transitioned to $I_h$ with a probability given by the parameter $p_h$, and transitioned to $I_l$ with a probability given by $(1 − p_h)$. The relative transmission rate of $I_h$ to $I_l$ individuals was set such that the highly infectious class ($I_h$) was responsible for 80% of secondary cases. As such, we were able to modify the extent of transmission heterogeneity by modifying $p_h$. A $p_h$ value of 0.8 implements a model with no transmission heterogeneity (as 80%

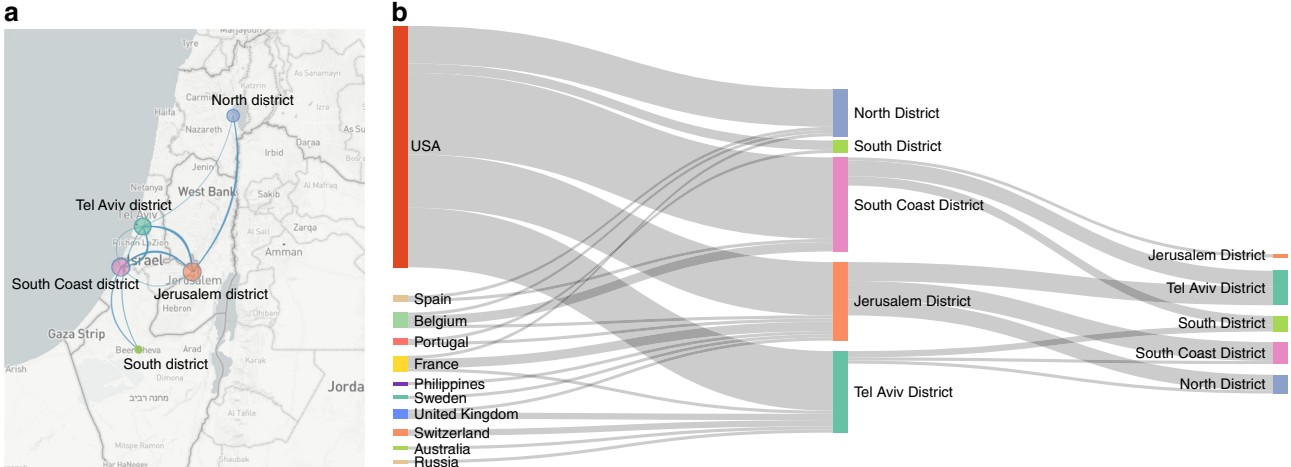

**Fig. 4 Spread of SARS-CoV-2 into and within Israel. a** Map of Israel with geographic locations of samples, and inferred spread inside Israel (blue lines). Figure generated using NextStrain[32]. **b** Inferred viral spread into and inside Israel, with directionality (left to right). Each line represents a transmission event inferred based on the phylogeny. Thicker lines indicate multiple transmission events. Source data are provided as a Source Data file.

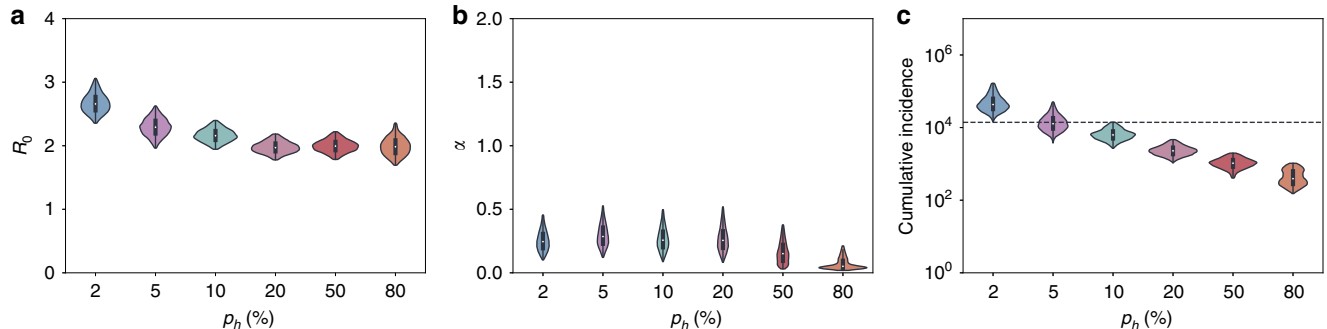

**Fig. 5 Estimated epidemiological parameters and cumulative incidence across different levels of transmission heterogeneity.** The parameter $p_h$ gives the fraction of infected individuals that are responsible for 80% of secondary infections. Higher $p_h$ values correspond to less transmission heterogeneity. **a** Estimated $R_0$ in Israel prior to March 19, 2020. **b** Estimated factor by which $R_0$ in Israel changed after March 19. **c** Estimated cumulative number of infected individuals in Israel on the date of the last sampled sequence (April 22, 2020). Horizontal dotted line at $N = 13,942$ shows the cumulative number of reported cases on April 22, 2020, as given by the ECDC (https://opendata.ecdc.europa.eu/covid19/casedistribution/csv). In **a–c**, only values that fall within the 95% highest posterior density intervals from the main MCMC chain are shown (total of 4751 data points). Violin plots show the kernel density estimation of the underlying distribution. The median value is denoted by a white dot and the black bar in the center of the violin defines the interquartile range. The black line stretched from the bar extends to the range of data that are not more than 1.5 times the interqaurtile range above the upper or below the lower quartile. Density is only plotted over the range of observed values. Results shown assume a time-varying migration rate estimated from a global maximum-likelihood phylogeny. Source data are provided as a Source Data file.

of the infected individuals are responsible for 80% of secondary infections), whereas a $p_h$ value of 0.2 implements a model consistent with the 20/80 superspreading rule[27] with 20% of individuals being responsible for 80% of secondary infections. Values of $p_h$ under 0.2 implement a model with even more extreme levels of transmission heterogeneity. The SEIR-type model we implemented further included terms for migration into and out of Israel; these terms enabled lineages in the phylogeny to transition out of Israel going backward in time (as would happen, in forward time, during an importation event). We considered two different functional forms for this migration term, one based on phylogenetically inferred timing of importations and the other assuming a simpler, constant rate of migration (Methods).

Using the migration rate form based on inferred importation times, we estimated $R_0$ prior to March 19 to be between 2.1 and 2.3 across the range of $p_h = 0.1$–0.8 (that is, 10–80%) with estimates increasing toward $R_0 = 3.0$ at high levels of super-spreading ($p_h = 2\%$) (Fig. 5a). Across the full range of $p_h = 2$–80%, we robustly estimated that quarantine measures had the effect of reducing $R_0$ by more than two-thirds ($\alpha = \sim 25\%$, where

$R_0$ following quarantine measures was given by $\alpha$ times $R_0$ prior to the implementation of these measures; Fig. 5b).

Figure 5c shows the cumulative number of SARS-CoV-2 cases by April 22, estimated by our phylodynamic analyses across the considered range of transmission heterogeneity. Estimates of the cumulative number of cases is highly sensitive to the level of assumed transmission heterogeneity, particularly at high levels of superspreading ($p_h = 2$–10%). Comparison between these inferred cumulative cases and reported case numbers (dotted lines in Fig. 5c) indicates that SARS-CoV-2 transmission dynamics were driven by a high level of viral superspreading. Specifically, if we assume almost complete case reporting, our phylodynamic analysis indicates that between 5% and 10% of infections are responsible for 80% of secondary infections. With lower assumed levels of case reporting, less than 5% of infections would be responsible for 80% of secondary infections. Findings from the phylodynamic analyses were shown to be robust to the specific functional form of the migration rate that was assumed, as well as to the overall magnitude of migration across a broad range of values (Supplementary Figs. 5 and 6).

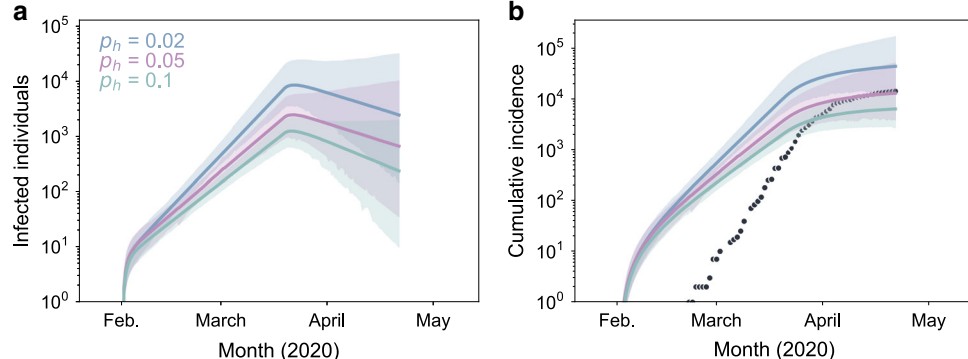

**Fig. 6 Epidemiological dynamics inferred using phylodynamic analysis. a** Estimated number of currently infected individuals ($I_l + I_h$) over time. **b** Estimated cumulative number of infected individuals. An infected individual is assumed to contribute to cumulative incidence at the end of their infectious period. Black dots show the cumulative number of reported cases in Israel over time. In **a** and **b**, lines show median estimates of models with different levels of transmission heterogeneity. Shaded regions represent the 95% highest posterior density region. Source data are provided as a Source Data file.

Phylodynamic analysis further allows us to visualize inferred epidemiological dynamics. In Fig. 6, we show inferred patterns of prevalence (Fig. 6a) and incidence (Fig. 6b) for three different assumed levels of viral superspreading. Inferred patterns of prevalence corroborate epidemiological findings that the number of cases started to decline in early April. Inferred patterns of cumulative incidence indicate that reporting rates were initially low but improved considerably over the time course of viral spread. The leveling off of cumulative incidence around late March/early April is observed in both the reported case data and in our inferred epidemiological dynamics, ground-truthing the results of our phylodynamic analyses.

Overall, our findings highlight the use of genomic data to effectively track the spread of an emerging virus using phylogenetic and phylodynamic approaches that have been developed to study viral outbreaks. We have found a relatively high proportion of genomes with short deletions, suggesting that such deletions arise frequently during SARS-CoV-2 replication and are to some extent tolerated by the virus. We succeeded in tracking the main transmission chains that led to SARS-CoV-2 spread in Israel, and applied phylodynamic analysis to infer the key epidemiological parameters governing its spread. Our results indicate that superspreading events drive the transmission dynamics of SARS-CoV-2, suggesting that focused measures to reduce contacts of select individuals/social events could mitigate viral spread. Finally, our results highlight how global connectivity allows for massive introductions of a virus and emphasize how border control and shelter-in-place restrictions are crucial for halting viral spread. Addendum September 2020: the authors would like to note that unfortunately, following the relaxation of social distance measures in May, case counts have substantially climbed and Israel has entered a second lockdown.

## Methods

**Ethics statement**. An exemption from institutional review board approval was determined by the Israeli Ministry of Health as part of an active epidemiological investigation, based on the use of retrospective anonymous data only and no medical intervention. This included exemption from informed consent. The study was further approved by the Tel-Aviv University ethics committee (approval 0001274-1).

**Details of samples and virus genome sequencing**. With the aim of generating a random sample of viral infections across the entire country, a total of 213 samples were retrieved from six major hospitals in Israel spanning the entire geography of Israel from south to north (Table 1 and Supplementary Table 2).

We obtained RNA extracted from nasopharyngeal samples. Sequencing was performed based on the V3 Artic protocol (https://artic.network/ncov-2019).

Briefly, reverse transcription and multiplex PCR of 109 amplicons was performed, and adapters were ligated to allow for sequencing. All samples were run on an Illumina Miseq using 250-cycle V2 kits in the Technion Genome Center (Israel). Supplementary Table 13 contains all primer names and sequences as described in the Artic protocol.

---

**Table 1 Summary of samples successfully sequenced.**

**a**

**Age group**

| Age group | Number of samples |
|---|---|
| 0–9 | 8 |
| 10–19 | 17 |
| 20–29 | 42 |
| 30–39 | 28 |
| 40–49 | 26 |
| 50–59 | 29 |
| 60–69 | 31 |
| 70–79 | 15 |
| 80–89 | 11 |
| 90 and up | 3 |
| Unknown | 2 |

**b**

**Location and hospital**

| Hospital | Geographic region | Number of samples |
|---|---|---|
| Barzilai Medical Center | South coast district | 30 |
| Samson Assuta Ashdod University Hospital | South coast district | 23 |
| Hadassah University Hospital - Ein Kerem | Jerusalem district | 62 |
| Poria Medical Center | North district | 26 |
| Sheba Medical Center | Tel-Aviv district | 51 |
| Soroka Medical Center | South district | 20 |

**c**

| Sex | Number of samples |
|---|---|
| Female | 101 |
| Male | 111 |

The table is divided by metadata information (a–c). Source data are provided as a Source Data file.

**Determining genome consensus sequences**. Sequencing reads were trimmed using pTrimmer, a multiplexing primer trimming tool[28], and then aligned to the reference genome of SARS-CoV-2 (GenBank ID MN908947) using our AccuNGS pipeline[29], which is based on BLAST[30], using an e-value of $10^{-9}$. The pipeline allows for consensus determination and variant calling. We considered substitutions at the consensus sequence (as compared to the reference) only if a given base was present in 80% of the aligned reads, and five or more reads aligned to the reference; bases where the majority of reads showed a substitution but that did not fulfill these two conditions were deemed uncertain. Similarly, positions to which no reads were mapped were also deemed uncertain, and such positions were assigned with an "N". All deletions were manually verified: (a) over 98% of the reads covering the deletion site mapped to both ends of the deletion (i.e., bore evidence of the deletion), (b) the deletion was based on over 40 independent reads (on average >1000 reads), and (c) coverage was high at both ends of the deleted region. Only sequences that spanned 90% of the reference genome were retained, leading to the removal of one sequence (Supplementary Table 2), and hence a new set of 212 Israeli sequences was generated here. Another two Israeli sequences already available on GISAID were added to the phylogenetic analysis, leading to a total of 214 sequences from Israel.

The collection dates of the 214 Israeli sequences used in our analysis ranged from February 23 through April 22, 2020. The number of sequences is thus ~1.5% of the total number of reported cases on April 22.

**Phylogenetic analyses**. All available full-length SARS-CoV-2 genomes from outside of Israel (a total of 16,403 sequences) were retrieved from GISAID on May 5, 2020. All sequences from a non-human host as well as sequences with incomplete sampling date (YYYY-MM or YYYY-MM-XX) or a high level of uncertainty (>10% ambiguous bases marked as N) were removed. All available sequences were then down-sampled to 4693 representative sequences across the globe using the latest build of NextStrain ncov pipeline[31,32] (https://github.com/nextstrain/ncov); 1195 of these 4693 sequences were from the U.S., while 1991 were from Europe. The 212 new Israeli sequences were added to the tree.

**Down-sampling of global tree for phylodynamic analysis**. Following the initial sampling of the global tree described above, we applied a second sampling specifically for the phylodynamic analysis. The down-sampling was inspired by the recommended guidelines described for SARS-CoV-2 (sarscov2phylodynamics.org), and thus we applied two sampling techniques:

(i) Random time stratified sampling—We sampled a total of 100 sequences from outside of Israel across $v = 5$ time intervals such that each time interval contained ~20 sequences.
(ii) Closest sequence match—Defining $S_{ISR}$ as the set of all sequences from Israel, we sample the exogenous set of sequences from the global tree with the minimal cophenetic distance between each Israeli sequence belonging to $S_{ISR}$ as based on the maximum-likelihood phylogeny. This results in sequences closely related to sequences from Israel to be included in the analysis.

We next manually curated the sequences from Israel to ensure they represent a random sample across Israel. To this end, we removed samples suspected to be from the same household, samples with consecutive identifiers, or identical samples with similar identifiers and similar dates. Only one sample from a given household was chosen randomly. This led to a removal of six sequences.

Following down-sampling and manual curation, a phylogenetic tree was inferred using the NextStrain pipeline[32]. The tree topology was validated as a legitimate representative of the global tree by performing 1000 random samples containing 373 sequences from the global tree. The Kendall–Colijn metric[33] was used to assess the distance between each random sample and the original tree, allowing us to create a null distribution. The $\lambda$ parameter, which determines the trade-off between topology and branch length, was set to zero, thus accounting for the tree topology alone. The significance of the topology of the down-sampled tree as compared to the global tree was thus obtained by comparing the Kendall–Colijn metric of the down-sampled tree to the null distribution ($P = 0.003$).

**Timing and distribution of importations**. Given a time-resolved global tree (after the initial down-sampling from GISAID database and before down-sampling for phylodynamic analysis; 4693 sequences total), we assigned a country to each internal node using NextStrain's maximum-likelihood ancestral state reconstruction. We defined an importation event into Israel as a transition from a non-Israeli node to an Israeli node. We then used the dates associated with the internal nodes to generate a distribution of importation dates, which was used to parameterize the phylodynamic migration rate (Supplementary Fig. 8), as described below. Moreover, we inferred that the first introductions to Israel occurred already in late January/early February, as further supported by data from epidemiological investigations.

The internal nodes and their associated dates were further used to infer the number of transmission chains that could have been prevented by arresting flights from the U.S. earlier. Out of a total of 214 clade importations from the U.S., 118 (55%) were inferred to have occurred between February 26 and March 9, 103

between March 1 and March 9 (48%), and 42 between March 4 and March 9 (20%).

**Confidence in numbers and fractions of importation events**. Confidence in the relative number of importation events from the U.S. vs. Europe was assessed using two measures of confidence intervals. These were aimed at testing whether the set of exogenous (non-Israeli) sequences was biased, or whether the set of Israeli sequences was biased. First, we generated 1000 samples of the exogenous sequences using a bootstrap approach: we sampled $N$ sequences with replacement, where $N$ is the number of exogenous sequences. We then determined the fraction of importation events into Israel for each set. Second, we similarly bootstrapped only the local (Israeli) sequences using a similar approach and assessed the fraction of importation events into Israel for each bootstrapped set. The reported confidence interval includes the lower bound and higher bound of both bootstrapping schemes. We describe below our approach for inferring the timing of importation events.

**Phylodynamic analysis**. Phylodynamic analyses were conducted using BEAST2 v2.6.2 (ref. [34]) and PhyDyn v1.3.6 (ref. [35]). An HKY substitution model with a lognormal prior for $\kappa$ with mean $\log(\kappa) = 1.0$ and standard deviation of $\log(\kappa) = 1.25$ was used. We assumed no sites to be invariant and used an exponential prior for $\gamma$ with a mean of 1.0. A strict molecular clock with a uniform prior between 0.0007 and 0.002 substitutions/site/year was used. A uniform prior was used for nucleotide frequencies. The down-sampled maximum-likelihood tree generated using IQ Tree was used as a starting tree.

PhyDyn is a coalescent-based inference approach implemented in BEAST2, allowing for the integration over phylogenetic uncertainty[35,36]. The program requires specification of an underlying epidemiological model, as well as any priors on parameters that will be estimated. In line with recent analyses[37], we assumed that the epidemiological dynamics of SARS-CoV-2 were governed by SEIR dynamics. Transmission heterogeneity has previously been described for viral pathogens including SARS-CoV-1 (ref. [38]) and appears to be important in the transmission dynamics of SARS-CoV-2 (refs. [25,26]). To account for the possibility of transmission heterogeneity, as in previous work[37], we modeled two classes of infected individuals: one with low transmissibility $I_l$ and one with high transmissibility $I_h$. Mathematically, the epidemiological model is given by Eqs. (1)–(5):

$$\frac{dS}{dt} = -\beta_l I_l \cdot \left(\frac{S}{N}\right) - \beta_h I_h \cdot \left(\frac{S}{N}\right) \tag{1}$$

$$\frac{dE}{dt} = \beta_l I_l \cdot \left(\frac{S}{N}\right) + \beta_h I_h \cdot \left(\frac{S}{N}\right) - \gamma_E E \tag{2}$$

$$\frac{dI_l}{dt} = (1 - p_h)\gamma_E E - \gamma_I I_l \tag{3}$$

$$\frac{dI_h}{dt} = p_h \gamma_E E - \gamma_I I_h \tag{4}$$

$$\frac{dR}{dt} = \gamma_I I_l + \gamma_I I_h. \tag{5}$$

We set as fixed the host population size to the population size of Israel, according to the European Centre for Disease Prevention and Control ($N = 8,883,800$), the average duration of time an individual spends in the exposed class ($1/\gamma_E = 3$ days), and the average duration of time an individual spends in the infected (infectious) class ($1/\gamma_I = 5.5$ days). These durations are based on a study that inferred transmissibility over the course of infection using data from established SARS-CoV-2 transmission pairs[39]. $R_0$ in this model is given by $(\beta_h p_h + \beta_l(1-p_h))/\gamma_I$, where $p_h$ is the fraction of exposed individuals who transition to the $I_h$ class instead of the $I_l$ class. In our model, we estimated a piecewise $R_0$ by estimating an initial $R_0$ that was in effect until March 19, 2020, when strong social distancing measures were implemented, along with a factor $\alpha$ by which $R_0$ changed on March 19.

Instead of independently parameterizing $\beta_h$ and $\beta_l$, we defined (as in previous work[37]) the relative transmissibility of infected individuals in the $I_h$ and $I_l$ classes by the parameter $\tau = \beta_h/\beta_l$, and simplify notation by defining $\beta \equiv \beta_l$. We further defined a parameter $P$ as the fraction of secondary infections that were caused by a fraction $p_h$ of the most transmissible infected individuals and set $P$ to 0.8. Based on set values of $P$ and $p_h$, we calculated $\tau$ as $(\frac{1-p_h}{p_h})/(\frac{1}{P} - 1)$. As such, we could easily parameterize the model across various levels of transmission heterogeneity, with a fraction $p_h$ of infected individuals being responsible for 80% of secondary infections. Existing epidemiological analyses indicate that $p_h$ is approximately 0.07–0.1 (7–10%)[25,26] indicative of even more transmission heterogeneity than given by the 20/80 rule[27]. We considered a range of $p_h$ between 2% and 80% in our phylodynamic analyses to allow for a broad range of transmission heterogeneity, from extreme superspreading ($p_h = 2$–10%) to no transmission heterogeneity ($p_h = 80\%$).

Again, based on existing analyses (sarscov2phylodynamics.org,[37]), we included an external reservoir in our analysis to allow for multiple introduced clades into

Israel to be jointly considered. Instead of modeling both exposed ($E$) and infected ($I$) individuals in the external reservoir, we assume a single infected class $Y$ undergoing exponential growth. We fix the duration of infection of this class of individuals to be 8.5 days ($= 1/\gamma_E + 1/\gamma_I$). We attempted to estimate both the growth rate of the infected class $Y$ and its initial size at time $t = 2019.7$, but found that these parameters were practically unidentifiable. We thus set the growth rate of the infected class to 24 person$^{-1}$ year$^{-1}$ (resulting in an $R_0$ of 1.56 in the reservoir) and estimated only the initial size of the infected class $Y$. An exponential prior with mean 1.0 was used for the initial size of $Y$.

Migration into and out of Israel occurred at an overall migration rate of $\eta(t)$ and we assumed for simplicity that all migrations involved exposed ($E$ class) individuals, rather than also implementing migration of the infected ($I_l$ and $I_h$) classes. As migration is assumed to be symmetrical into and out of Israel, it does not affect the focal SEIR model dynamics. However, it does influence the probability that a given lineage's geographic state is assigned to Israel. We considered two different functional forms of the migration rate $\eta(t)$. The first form for $\eta(t)$ was generated using the timing of inferred importation events (described above in "Timing and distribution of importations"). Inferred importation events were grouped into 3-day windows and a piecewise exponential function fit to the data using the Nelder–Mead algorithm as implemented in SciPy[40]. The curve was fixed to change from growth to decay at the end of the time window with the peak number of importations (Supplementary Fig. 9A) and assumed to be 0 until the date at which the best fit curve was ≥1. Model fitting resulted in an importation rate given by the curve $\exp(57(t - 2020.05))$ between 2020.05 and 2020.18 and declined from 2020.18 with the curve $1680\exp(-52(t - 2020.18))$. To assess the robustness of our results to the magnitude of this curve, we also scaled the growth and decay rates (57 and −52, respectively) by $\theta = 0.8, 0.9, 1.0, 1.1$, and 1.2 and modified the initial value of the importation rate at 2020.18 accordingly (Supplementary Figs. 5 and 9B and Supplementary Tables 3–7). Over the time series in our model, this translates to a total of 17, 33, 62, 118, and 228 migrations into and out of Israel that result in established clades. The second form for $\eta(t)$ assumed a constant migration rate ($\eta = 10, 100, 1000, 2500, 5000$ year$^{-1}$) (Supplementary Fig. 5 and Supplementary Tables 8–12) which translates to 6, 61, 607, 1518, and 3037 migrations over the time course of our model.

With a given functional form for $\eta(t)$ and with the parameterization of this form, we used our parameter estimates to infer the time-varying probability that an exposed individual in Israel migrated into Israel versus became infected locally (Supplementary Figs. 10 and 11). This time-varying probability is given by $\eta(t)/\left(\eta(t) + \frac{\beta S(I_l + \tau I_h)}{N}\right)$. At high migration rates ($\theta = 1.2, \eta \geq 2500$), this probability remained unrealistically high throughout the epidemic. At reasonable values of $\theta$ and $\eta$ (0.9–1.1 and 100–1000, respectively), the parameter estimates were relatively insensitive to $\eta$ (Supplementary Figs. 5 and 6). PhyDyn simultaneously estimates model parameters and phylogenies and thus integrates over phylogenetic uncertainty. Our extensive sensitivity analysis involving the importation/exportation rate $\eta$ was done to ensure the robustness of our inferred epidemiological parameters. The choice and magnitude of $\eta$ will affect not only the probability that viral lineages are inside or outside of Israel but the overall topology of the phylogenies and thus has the potential to impact epidemiological parameter estimation.

Our prior on $\alpha$, the factor by which $R_0$ changes on March 19, was a uniform prior between 0 and 2, thereby allowing $R_0$ to either increase, decrease, or remain unchanged after March 19. $I_l$ and $I_h$ were assumed to be negligibly small (1E−8) at the beginning of the SEIR dynamics. The PhyDyn $t_0$ parameter was set to 2019.7 and a constant population size coalescent model ($N_e = 0.1$) was used prior to this date when proposed trees had earlier root dates. SEIR dynamics were assumed to begin on February 1. Sequences sampled from Israel were randomly assigned to $I_h$ with probability $p_h$ and to $I_l$ with probability $1 - p_h$. XML files to run both BEAST2 and PhyDyn were generated using a custom Python 3 script, which was designed to edit a template XML file originally generated with BEAUti and manually edited. To aid in mixing, we used Adaptive Metropolis Coupled MCMC[41] with three chains. All MCMC chains were run for 10 million steps and convergence was assessed based on visual inspection of parameter traces. The first 50% of MCMC steps were discarded as burn-in. Maximum clade credibility trees were generated using TreeAnnotator. BEAST2 and PhyDyn outputs were visualized using Python 3, Matplotlib[42], Seaborn, and Baltic (https://github.com/evogytis/baltic). The ggtree package was used to visualize the Israeli phylogeny[43]. Supplementary Fig. 7 shows the time-aligned maximum clade credibility phylogeny of the Israeli sequences along with outside-Israel sequences for the model results shown in Figs. 5 and 6.

**Reporting summary**. Further information on research design is available in the Nature Research Reporting Summary linked to this article.

## Data availability
Data that support the findings of this study have been deposited in the relevant databases: a viral sequence per each patient sample was deposited in the GISAID database (https://www.gisaid.org) with accession numbers EPI_ISL_447258 - EPI_ISL_447469. The raw sequencing reads were deposited in the NCBI Sequence Read Archive (SRA) database under BioProject accession number PRJNA647529. The reference genome of SARS-CoV-2, ID MN908947, was downloaded from GenBank (https://www.ncbi.nlm.nih.gov/genbank/). A list of all sequence accession numbers used in this study, Beast XML configurations, and outputs are available at https://github.com/SternLabTAU/SARSCOV2NGS. Source data are provided with this paper.

## Code availability
All analysis scripts, NextStrain make file and phylodynamic model configuration scripts are available at https://github.com/SternLabTAU/SARSCOV2NGS. Our local NextStrain build is also available in NextStrain community at https://nextstrain.org/community/SternLabTAU/SARSCOV2NGS?f_country=Israel.

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

## Acknowledgements
We wish to thank Dr. Erik Volz for helpful discussions, as well as Dr. Boaz Lev at the Israeli Ministry of Health, Dr. Tal Katz-Ezov at the Technion Genome Center, and Stern lab members for their support during an ongoing pandemic and various stages of lockdown. This work was funded by the Israeli Science Foundation (1333/16), by the NIAID Centers of Excellence for Influenza Research and Surveillance (CEIRS) grant HHSN272201400004C, and by a generous donation from the Milner foundation and from AppsFlyer. This study was supported in part by a fellowship to D.M., T.K., and O.T. from the Edmond J. Safra Center for Bioinformatics at Tel-Aviv University. The phylodynamic analysis presented as part of this work used the High Performance Computing Environment provided by the MIDAS Coordination Center, supported by NIGMS grant 5U24GM132013 and the NIH STRIDES program.

## Author contributions
D.M., M.A.M., N.H., O.T., and T.K. performed all the analyses and drafted the paper. M.M. coordinated the sample sequencing. N.S., S.G.H., S.A., O.V., A.Sh., D.W., A.P., Y.S.A., and D.R.K. contributed clinical samples. N.M.K., A.H., and A.S. conceived and coordinated the study. A.S. and K.K. supervised and led the study.

## Competing interests
The authors declare no competing interests.
