## [Peer Review File · Nature Communications]

REVIEWER COMMENTS

Reviewer #1 (Remarks to the Author):

This is an interesting paper using the latest phylodynamic techniques as applied to the introduction and spread of SARS-CoV-2 in Israel. The techniques used have a lot of technical caveats (regarding sampling bias and model selection) which are reasonably indicated in the methods section, but if space allows it would be good to re-iterate some of these in the concluding paragraph (as non-tech as possible).

The paper is well written, and below are some minor detailed comments, questions and suggestions

SNVs and nucleotide differences -

Although only 200 or so genomes were analysed (there are now >50,000 genomes on GISAID) this section is good and describes what was observed well.

However, since the time of writing and the time of reviewing (many more genomes now available), it would be really interesting to know whether any of the deletions observed in the Israel samples have now been found elsewhere (table associated with figure 2); the CoV-GLUE <http://cov-glue.cvr.gla.ac.uk/#/home> or similar website might help? [I'm not suggesting that you perform a comprehensive global analysis, but some indication as to the common-ness or not of the deletions in other countries would be useful, otherwise you might just think that it was sequencing or platform error, or choice of primers or something]

Origins and transmission patterns in Israel -

The phylogenetic method for assessing the origin of the imports to Israel, whilst not perfect, is well explained and the consideration of sampling bias (many more sequences from US and Europe than Israel) is handled adequately.

It is an interesting observation that it seems that the US travellers contributed more to the importation of the virus than their raw numbers would suggest. 27% (raw numbers) vs 70% (clade importations) - the 70% is influenced by the tree structure and the country-trait model; since this is likely to be a contentious number I think it would be useful if you put in the text (bottom of page 5) some indication of confidence in this result (you have the boot strap values for each imported clade, and noted in the paragraph above about the testing with respect to tree structure, but this is buried in the methods and another sentence or so would be good here).

Also, probably add a sentence to explain where the 30-40% of transmission chains comes from - presumably this is because 30-40% of the onward transmission chains originated from those US imports, but you've not stated the number of chains nor where they are all from in the main text yet (bottom of page 5).

Phylodynamic modelling -

The phylodynamic SEIR model with transmission heterogeneity you have chosen to use here seems appropriate, and I think it is good that you are using these newer BEAST2 models to make the R_0 (R_e) estimates. As these models are quite new so don't have a long history it is useful to explain their basis (the methods is useful here). From just reading the results text it is not exactly clear why $ph=80\%$ means no heterogeneity (why is this not 100%? And can you give another descriptor which goes from 0-1 as well as the ph parameter?). Also, it is not quite clear from the results text, whether this is a model with 2 x SEIR; one compartment stream of normal spreaders and one of super spreaders (the proportion of which is given by ph), or whether it is just one SEIR with individuals with a long tailed distribution shape of heterogenous transmission parameters. On reading the detailed methods I see it is the former - but please add a sentence or so in the results text to clarify. For example - is this 2 x SEIR a bit like a age structured model? and/or do you

have the data to know if one tier is one type of population or not, e.g. community vs health care workers ?

Reviewer #2 (Remarks to the Author):

The manuscript presents an a genomic epidemiology analysis of Israeli SARS-CoV-2 sequences. SNPs and deletions in the genome are described, and the authors then proceed to perform a phylodynamics analysis to identify, (amongst others) the proportion of lineage importations coming from outside Israel, and the level of superspreading. This is a strong and thorough piece using state-of-the-art methods and I have no reservations in recommending publication.

The one major reservation I had concerns the deletions. The identification of two sequences with frameshift mutations, which are the only examples presented that possess these and are very close to each other in the global consensus, in a clade containing two other sequences with identified deletions, raises concerns about potential artefacts. The authors go to admirable lengths in trying to rule out sequencing error (p10), but there are other points in the process where something could go wrong. Were these samples collected by the same entity? Are they geographically linked? Do closely-related examples from the global alignment exhibit similar phenomena?

I also have a couple of queries about the ν parameter. Firstly, if it "does not affect the focal SEIR model dynamics" (p14), I'm a bit uncertain why varying it should affect estimates of R_0 or α at all. It is possible that there is a complexity of the model here that I am missing, but referring to Volz, Fu et al. has not helped. Further, it is assumed to be constant despite the travel ban, at which point presumably the number of external importations drastically decreased. Could the model accommodate a change in this at the point that travel was suspended?

Minor comments:

* It is not entirely clear to me what the ancestral state reconstruction procedure for locations was. Is it the Nextstrain routine (which I believe is maximum likelihood) or the parsimony approach of Volz, Boyd et al.?

* In the "rate of importations" section, the text "mid-branch date for each node leading to an Israeli tip" should surely be "mid-branch date for each node leading to an Israeli node".

Matthew Hall

Response to reviewer comments

Reviewer #1 (Remarks to the Author):

This is an interesting paper using the latest phylodynamic techniques as applied to the introduction and spread of SARS-CoV-2 in Israel. The techniques used have a lot of technical caveats (regarding sampling bias and model selection) which are reasonably indicated in the methods section, but if space allows it would be good to re-iterate some of these in the concluding paragraph (as non-tech as possible).

The paper is well written, and below are some minor detailed comments, questions and suggestions.

We thank the reviewer for the positive feedback.

Comment 1: SNVs and nucleotide differences - Although only 200 or so genomes were analysed (there are now >50,000 genomes on GISAID) this section is good and describes what was observed well.

However, since the time of writing and the time of reviewing (many more genomes now available), it would be really interesting to know whether any of the deletions observed in the Israel samples have now been found elsewhere (table associated with figure 2); the CoV-GLUE <http://cov-glue.cvr.gla.ac.uk/#/home> or similar website might help ? [I'm not suggesting that you perform a comprehensive global analysis, but some indication as to the common-ness or not of the deletions in other countries would be useful, otherwise you might just think that it was sequencing or platform error, or choice of primers or something]

Response 1:

We thank the reviewer for this important comment. We have now searched for the deletions we detected in >50,000 sequences available in GISAID. Indeed, some are present and even quite frequent in samples spanning the globe. We note that this may be an underestimate as deletions may be masked (for example in the CoV-GLUE site). To address this comment, we have added an additional column to Figure 2 (shown below) that presents the number of non-Israeli samples that carry the same deletion as observed in our samples.

#	Genome sites	Length (nt)	ORF/Genomic location	Putative effect	# samples found in	Sample IDs	Number of reads supporting deletion	Geographic location (district)	# non-Israeli samples found in ^{1,2}
1	686-694	9	ORF1ab polyprotein	Deletion of 3 amino acids	2	2086008, 130710157	3575, 1852	Jerusalem, Tel Aviv	222
2	3882-3899	18	ORF1ab polyprotein	Deletion of 6 amino acids and an additional single amino acid mutation	2	2089839, 2089852	427, 605	Jerusalem	--
3	27387-27396	10	End of ORF6 and start of ORF7a	Stop codon of ORF6 is recreated. Start codon of ORF7a is deleted with no in-frame replacement	1	13077726	3801	Tel Aviv	--
4	28254	1	End of ORF8	Last amino acid is replaced by a 5 amino acid addition	1	2086033	2849	Jerusalem	15
5	29746-29748	3	3' UTR	Non-coding, unknown	2	51137844, 51141225	42,147	South	--

¹ Based on >50,000 SARS-CoV-2 sequences downloaded from GISAID, July 17 2020, and also reported in <https://virological.org/t/common-microdeletions-in-sars-cov-2-sequences/485>

² We note that these are underestimates as deletions are sometimes masked in reported sequences.

Origins and transmission patterns in Israel -

The phylogenetic method for assessing the origin of the imports to Israel, whilst not perfect, is well explained and the consideration of sampling bias (many more sequences from US and Europe than Israel) is handled adequately.

Thanks!

Comment 2: It is an interesting observation that it seems that the US travellers contributed more to the importation of the virus than their raw numbers would suggest. 27% (raw numbers) vs 70% (clade importations) - the 70% is influenced by the tree structure and the country-trait model; since this is likely to be a contentious number I think it would be useful if you put in the text (bottom of page 5) some indication of confidence in this result (you have the boot strap values for each imported clade, and noted in the paragraph above about the testing with respect to tree structure, but this is buried in the methods and another sentence or so would be good here).

Response 2: We agree with the reviewer, and we now highlight at the top of page 6:

"There is a strong discrepancy between this 27% estimate and the 70% estimate for clade introductions, and this discrepancy holds even when considering our lower bound bootstrap estimate of 50% for clade importations (mentioned above). This suggests that the travelers returning from the U.S. contributed substantially more to the spread of the virus in Israel than would be proportionally expected."

Comment 3: Also, probably add a sentence to explain where the 30-40% of transmission chains comes from - presumably this is because 30-40% of the onward transmission chains originated from those US imports, but you've not stated the number of chains nor where they are all from in the main text yet (bottom of page 5).

Response 3:

We have now added the explicit values on which these percentages are based, and we have further elaborated in the Methods section how these values were inferred (indeed based on the onwards transmission chains). We have simplified the "Rate of importation" section and it is now named "Timing and distribution of importations". We now write in the text in the results section (top of page 6):

"Moreover, by examining the timing of viral importations events from the U.S. into Israel, we found that up to 55% of the transmission chains in Israel (118 out of 214; Methods) could have been prevented had flights from the U.S. been arrested at the same time that flights from Europe were arrested (between February 26 and March 4, instead of by March 9)."

And in the Methods section:

"Given a time resolved global tree (after the initial down sampling from GISAID database and before down-sampling for phylodynamic analysis; 4,693 sequences total), we assigned a country to each internal node using NextStrain's maximum-likelihood ancestral state reconstruction. We defined an importation event into Israel as a transition from a non-Israeli node to an Israeli node. We then used the dates associated with the internal nodes to generate a distribution of importation dates, which was used to parameterize the phylodynamic migration rate (Fig. S8), as described below. Moreover, we inferred that the first introductions to Israel occurred already in late January/early February, as further supported by data from epidemiological investigations.

The internal nodes and their associated dates were further used to infer the number of transmission chains that could be prevented by arresting flights from the U.S. earlier. Out of a total of 214 clade importations from the U.S., 118 (55%) were inferred to have occurred between Feb. 26 and March 9, 103 between March 1 and March 9 (48%), and 42 between March 4 and March 9 (20%)."

Comment 4:

Phylodynamic modelling -

The phylodynamic SEIR model with transmission heterogeneity you have chosen to use here seems appropriate, and I think it is good that you are using these newer BEAST2 models to make the R_0 (R_e) estimates. As these models are quite new so don't have a long history it is useful to explain their basis (the methods is useful here). From just reading the results text it is not exactly clear why $ph=80\%$ means no heterogeneity (why is this not 100%? And can you give another descriptor which goes from 0-1 as well as the ph parameter?). Also, it is not quite clear from the results text, whether this is a model with 2 x SEIR; one compartment stream of normal spreaders and one of super spreaders (the proportion of which is given by ph), or whether it is just one SEIR with individuals with a long tailed distribution shape of heterogenous transmission parameters. On reading the detailed methods I see it is the former - but please add a sentence or so in the results text to clarify.

For example - is this 2 x SEIR a bit like an age structured model? and/or do you have the data to know if one tier is one type of population or not, e.g. community vs health care workers?

Response 4:

We thank the reviewer for this comment and have edited text in order to clarify the structure of the phylodynamic model. Specifically, we have added text to lines 251-258 (results) as well as lines 497-507 (methods) which explain why a $p_h = 0.80$ represents no transmission heterogeneity. In short, because we fix the proportion of infections to be caused by individuals in the high infectious class to be 80% ($P = 0.80$), when 80% of the infectious individuals end up in this class ($p_h = 0.80$) there is no superspreading: 80% of the infections are caused by 80% of the infectious individuals. Note also that the value of parameter τ , which specifies the factor by which high infectious individuals are more infectious than low infectious individuals, evaluates to 1 when $p_h = 0.8$, indicating that under this parameterization, high infectious individuals are just as infectious as low infectious individuals and there is no transmission heterogeneity. Because we have clarified the text, we have chosen not to introduce another parameter to quantify the degree of superspreading.

Additionally, we have added a supplemental figure (Fig. S4) showing the structure of the epidemiological model. We chose to model two infectious groups (with high and low infectiousness) rather than an entire distribution of infectiousness because for the coalescent process it is only necessary to specify the mean and the variance of the offspring distribution, rather than the entire distribution.

While we let infection dynamics within Israel be governed by SEIR-type dynamics, we had to include an exogenous compartment to allow lineages to come into and out of Israel. The dynamics within this exogenous compartment are assumed to be governed by exponential growth.

Unfortunately, the data included in the present study are anonymized. Therefore, we do not have the epidemiological data that would be needed to gauge which factors or attributes may be associated with higher infectiousness.

Reviewer #2 (Remarks to the Author):

The manuscript presents a genomic epidemiology analysis of Israeli SARS-CoV-2 sequences. SNPs and deletions in the genome are described, and the authors then proceed to perform a phylodynamics analysis to identify, (amongst others) the proportion of lineage importations coming from outside Israel, and the level of superspreading. This is a strong and thorough piece using state-of-the-art methods and I have no reservations in recommending publication.

We thank the reviewer for this positive feedback.

Comment 1: The one major reservation I had concerns the deletions. The identification of two sequences with frameshift mutations, which are the only examples presented that possess these and are very close to each other in the global consensus, in a clade containing two other sequences with identified deletions, raises concerns about potential artefacts. The authors go to admirable lengths in trying to rule out sequencing error (p10), but there are other points in the process where something could go wrong. Were these samples collected by the same entity? Are they geographically linked? Do closely-related examples from the global alignment exhibit similar phenomena?

Response 1: We thank the reviewer for this important comment. The samples with deletions stem from different medical centers, different geographical regions, and were also run in different sequencing batches. In response to reviewer 1 (see above), we have extended our analysis by assessing whether the deletions we observed also appear in other available (non-Israeli) GISAID sequences. Some do – please see our revised Figure 2. We have edited text in the manuscript to describe these findings (top of page 5):

“When focusing on deletions that occurred in two samples, we noted that deletion #5 was present in two related samples that were sampled five days apart from each other. Deletion #1, on the other hand, appeared in two samples located in very remote clades of the phylogeny. This deletion has been observed multiple times in various sequences with diverse genetic backgrounds, including sequences from many different countries across the world, suggesting that it has arisen multiple independent times (Fig, 2B). Deletions #2, #3 and #4 revealed an intriguing pattern: three independent deletions (one of which was present in two samples) were all part of the same clade that included eighteen samples (Fig. 2). One non-synonymous SNV defined this clade: S2430R in ORF1b, which affects the non-structural protein NSP16. This protein has been reported to be a 2’O-methyltransferase that enhances evasion of the innate immune system (Menachery, et al. 2014). To follow up on our finding of deletions in this clade, we examined a set of over 50,000 global sequences, and found that 137 sequences likely belong to the clade defined by S2340R, with eight of these sequences bearing short deletions (Table S1). Interestingly, we found that the proportion of these unique deletions observed in the S2340R-defined clade was 5%, significantly higher than the proportion of unique deletions observed across the entire global tree

(1.8%) (P=0.01; hypergeometric test), leading us to cautiously suggest that S2340R is associated with a higher rate of deletions”.

Comment 2: I also have a couple of queries about the η parameter. Firstly, if it "does not affect the focal SEIR model dynamics" (p14), I'm a bit uncertain why varying it should affect estimates of R_0 or α at all. It is possible that there is a complexity of the model here that I am missing, but referring to Volz, Fu et al. has not helped. Further, it is assumed to be constant despite the travel ban, at which point presumably the number of external importations drastically decreased. Could the model accommodate a change in this at the point that travel was suspended?

Response 2:

As pointed out by the reviewer, because the η parameter is balanced in and out of Israel in our model it does not directly influence the modeled dynamics. However, because the BEAST likelihood calculations incorporate both the likelihood associated with the model of sequence evolution (clock rate, etc.) as well as the likelihood of the PhyDyn model, the η parameter can also influence the inferred tree topology, which can in turn influence the inferred epidemiological parameters. For example, setting a low η parameter will force the tree structure to group sequences in the focal region more closely in the tree to limit the number of state transitions into the focal region. This has been clarified in the Methods section with the text (lines 523-524):

“However, it does influence the probability that a given lineage’s geographic state is assigned to Israel.”

As well as with the text (lines 551-553):

“The choice and magnitude of η will affect not only the probability that viral lineages are inside or outside of Israel but the overall topology of the phylogenies and thus has the potential to impact epidemiological parameter estimation. “

In response to the reviewers point about non-constant import rates, we have incorporated an analysis into the main text which uses the ancestral state reconstruction and inferred transitions into Israel from a large (~5000 sequences) global phylogeny to infer introduction dates into Israel. A piecewise exponential curve has been fit to these data and incorporated into our phylodynamic model as a time varying η parameter. To assess sensitivity to the magnitude of this curve we have scaled the growth and decay rates of the fit exponential curves by factors ranging from 0.8 to 1.2. Our results are robust to these variations in the η parameter. This analysis has been described in the methods section with the text:

“Given a time resolved global tree (after the initial down sampling from GISAID database and before down-sampling for phylodynamic analysis; 4,693 sequences total), we assigned a country to each internal node using NextStrain’s maximum-likelihood ancestral state reconstruction. We defined an importation event into Israel as a transition from a non-Israeli node to an Israeli node. We then used the dates associated with the internal nodes to generate a distribution of importation dates, which was used to parameterize the phylodynamic migration rate (Fig. S8), as described below. “

“The first form for $\eta(t)$ was generated using the timing of inferred importation events (described above in Timing and distribution of importations). Inferred importation events were grouped into three-day windows and a piecewise exponential function fit to the data using the Nelder-Mead algorithm as implemented in SciPy (Virtanen et al., 2020). The curve was fixed to change from growth to decay at the end of the time window with the peak number of importations (Fig S9A) and assumed to be 0 until the date at which the best fit curve was ≥ 1 . Model fitting resulted in an importation rate given by the curve $\exp(57(t - 2020.05))$ between 2020.05 and 2020.18 and declined from 2020.18 with the curve $1680 \exp(-52(t - 2020.18))$. To assess the robustness of our results to the magnitude of this curve, we also scaled the growth and decay rates (57 and -52, respectively) by $\theta = 0.8, 0.9, 1.0, 1.1, \text{ and } 1.2$ and modified the initial value of the importation rate at 2020.18 accordingly (Fig. S9B, Fig. S5, Tables S3-S7). Over the time series in our model, this translates to a total of 17, 33, 62, 118, and 228 migrations into and out of Israel that result in established clades. “

Minor comments:

* It is not entirely clear to me what the ancestral state reconstruction procedure for locations was. Is it the Nextstrain routine (which I believe is maximum likelihood) or the parsimony approach of Volz, Boyd et al.?

- We used the Nextstrain routine, which implements ML-based ancestral state reconstruction using TreeTime. We have clarified this in the text.

* In the "rate of importations" section, the text "mid-branch date for each node leading to an Israeli tip" should surely be "mid-branch date for each node leading to an Israeli node".

- Many thanks for noticing this. We have fixed this in the text with some clarifications.

REVIEWERS' COMMENTS

Reviewer #1 (Remarks to the Author):

Thanks for considering my comments; and your response and changes to the manuscript look OK.

Specifically

Comment 1 (SNPs and indels)

Thanks for including the additional column in the table of figure 2 - I think this helps show that at least some of the deletions are seen elsewhere. Also the edited text (top of page 5) is OK too.

Comment 2 (27% vs 70%) & Comment 3 (number of chains)

These additions are good.

Comment 4 (Re and ph)

Thanks for clarifying about the ph - this is now fine.

Also, the changes regarding time varying importation rates look to be OK as well.

Reviewer #2 (Remarks to the Author):

My comments have been dealt with. Congratulations to the authors for a very strong piece.